# Comparison of the Therapeutic Effects of Adipose- and Bone Marrow-Derived Mesenchymal Stem Cells on Renal Fibrosis

**DOI:** 10.3390/ijms242316920

**Published:** 2023-11-29

**Authors:** Maria Yoshida, Ayumu Nakashima, Naoki Ishiuchi, Kisho Miyasako, Keisuke Morimoto, Yoshiki Tanaka, Kensuke Sasaki, Satoshi Maeda, Takao Masaki

**Affiliations:** 1Department of Nephrology, Hiroshima University Hospital, 1-2-3 Kasumi, Minami-ku, Hiroshima 734-8551, Japan; mariayoshida@hiroshima-u.ac.jp (M.Y.); ishiuchi@hiroshima-u.ac.jp (N.I.); kishomiyasako@gmail.com (K.M.); h01.morimori.ksk.0221@gmail.com (K.M.); zebra.y.t.06@gmail.com (Y.T.); kensasaki@hiroshima-u.ac.jp (K.S.); 2Department of Stem Cell Biology and Medicine, Graduate School of Biomedical & Health Sciences, Hiroshima University, 1-2-3 Kasumi, Minami-ku, Hiroshima 734-8553, Japan; s-maeda@twocells.com; 3TWOCELLS Company, Limited, 16-35 Hijiyama-honmachi, Minami-ku, Hiroshima 732-0816, Japan

**Keywords:** mesenchymal stem cell, renal fibrosis, adipose tissue, bone marrow

## Abstract

Mesenchymal stem cells (MSCs) have attracted a great deal of interest as a therapeutic tool for renal fibrosis. Although both adipose-derived and bone marrow-derived MSCs (ADSCs and BMSCs, respectively) suppress renal fibrosis, which of these two has a stronger therapeutic effect remains unclear. This study aimed to compare the antifibrotic effects of ADSCs and BMSCs extracted from adipose tissue and bone marrow derived from the same rats. When cultured in serum-containing medium, ADSCs had a more potent inhibitory effect than BMSCs on renal fibrosis induced by ischemia-reperfusion injury in rats. ADSCs and BMSCs cultured in serum-free medium were equally effective in suppressing renal fibrosis. Mice infused with ADSCs (serum-containing or serum-free cultivation) had a higher death rate from pulmonary embolism than those infused with BMSCs. In vitro, mRNA levels of tissue factor, tumor necrosis factor-α-induced protein 6 and prostaglandin E synthase were higher in ADSCs than in BMSCs, while that of vascular endothelial growth factor was higher in BMSCs than in ADSCs. Although ADSCs had a stronger antifibrotic effect, these findings support the consideration of thromboembolism risk in clinical applications. Our results emphasize the importance of deciding between ADSCs and BMSCs based upon the target disease and culture method.

## 1. Introduction

Chronic kidney disease (CKD) affects 8–16% of the general population worldwide [1], resulting in decreased quality of life, increased economic burden and increased mortality rate. To alleviate the progression of CKD, several pharmacological treatments, such as renin–angiotensin–aldosterone system blockades [2] and sodium–glucose co-transporter 2 inhibitors [3,4], have been developed. However, the beneficial effects of these medications are minimal; thus, CKD eventually progresses to end-stage kidney disease in many patients. New, more effective therapies for alleviating CKD are required.

Mesenchymal stem cells (MSCs) are multipotent adult stem cells that accelerate the repair of injured tissues through paracrine-mediated actions [5]. MSCs can be obtained from various tissues, such as bone marrow, adipose, dental pulp and umbilical cord tissues. MSCs derived from bone marrow (BMSCs) and adipose tissue (ADSCs) are frequently used because of their relative availability. According to previous studies, transplantations of BMSCs [6,7,8,9,10] and ADSCs [11,12] have each alleviated renal fibrosis in various models. It has been reported that the injection of BMSCs promotes tubular regeneration in a cisplatin-treated model [6,7] through the release of insulin-like growth factor-1 [7], resulting in improvement of renal function. Other studies have shown that the administration of BMSCs suppresses renal fibrosis in the IRI model [8], by decreasing the expression of pro-inflammatory cytokines [9]. Furthermore, BMSCs’ transplantation has been shown to improve renal function in an IRI model of immunosuppressed rats concurrently with decreased matrix metalloproteinase-2 activity [10]. On the other hand, the injection of ADSCs has been reported to reduce kidney injury and renal fibrosis through Sox9 activation in the IRI model [11], and to reduce urinary protein and fibrosis markers in a renal artery stenosis model [12]. From these studies, there is great promise for their clinical application in kidney disease. However, which of these two types has a stronger attenuation effect on renal fibrosis remains unclear.

Here, we aimed to directly compare the antifibrotic effects of ADSCs and BMSCs on renal fibrosis induced by ischemia-reperfusion injury (IRI) in rats, and to provide evidence to support the selection of appropriate MSCs for the treatment of renal disease. Our previous study showed that MSCs cultured in serum-free medium enhanced the gene expression of tumor necrosis factor-α-induced protein 6 (TSG-6), resulting in the greater amelioration of renal fibrosis than MSCs cultured in serum-containing medium [13]. Therefore, we also compared the anti-fibrotic effect of ADSCs and BMSCs cultured in serum-free medium in addition to serum-containing medium.

In this study, we showed that ADSCs cultured in serum-containing medium had a more robust effect on the attenuation of renal fibrosis in IRI-model rats than BMSCs cultured in the same medium. By contrast, ADSCs and BMSCs cultured in serum-free medium had an equivalent inhibitory effect on renal fibrosis. Furthermore, we demonstrated that thrombosis is an important issue in the clinical application of ADSCs. In in vitro experiments, we revealed that the mRNA level of tissue factor (TF) was higher in ADSCs than in BMSCs. Furthermore, the mRNA levels of TSG-6 and prostaglandin E synthase (PTGES) were higher in ADSCs than in BMSCs, while that of vascular endothelial growth factor (VEGF) was higher in BMSCs than in ADSCs. Our data provide further evidence that the choice between BMSCs and ADSCs should be based on the target disease and the culture method.

## 2. Results

### 2.1. Comparison of the Therapeutic Effects of ADSCs and BMSCs Cultured in Serum-Containing Medium on IRI-Induced Renal Fibrosis in Rats

First, we compared the therapeutic effects of intravascular ADSCs and BMSCs cultured in Dulbecco’s modified Eagle’s medium (DMEM) with 10% fetal bovine serum (FBS) (a type of basal medium commonly used for the cultivation of MSCs) on renal fibrosis. We examined the expression of anti-alpha-smooth muscle actin (α-SMA) and transforming growth factor (TGF)-β1 in IRI model rats infused with PBS, ADSCs or BMSCs at 21 days post-IRI. As shown in Figure 1a, the protein levels of α-SMA and TGF-β1 were remarkably increased in IRI rats infused with phosphate-buffered saline (PBS) compared with those in the sham group. These increases were significantly suppressed in IRI rats infused with ADSCs, but not in the IRI rats infused with BMSCs, which showed no significant differences with the PBS group. Immunostaining revealed that the α-SMA-, collagen I- and collagen III-positive areas were reduced in the ADSC group compared with those in the BMSC group (Figure 1b,c). Similarly, hematoxylin and eosin (HE) and Masson’s trichrome staining revealed that the tubulointerstitial injury score and the area of interstitial fibrosis were more significantly reduced in the ADSC group than in the BMSC group (Figure 1d,e). These findings indicated that ADSCs prevented renal fibrosis more effectively than BMSCs in IRI rats.

### 2.2. Comparison of Tail Vein Infusion of ADSCs and BMSCs on the Incidence of Death in Mice

The procoagulant activity of MSCs raises the concern of thrombogenic risk [14]. A previous study reported that the high-dose infusion of mouse ADSCs (1.0 × 10^5^ cells per mouse) cultured in serum-containing medium induced pulmonary embolism and a high mortality rate [15]. Therefore, we investigated the risk of pulmonary embolism induced by high-dose infusion of ADSCs and BMSCs cultured in serum-containing medium into the tail vein of C57/BL6 mice. As shown in Table 1, all of the mice (n = 10/10) that received intravenous infusion of PBS and 1.0 × 10^5^ BMSCs were alive 24 h later. By contrast, 40% (n = 4/10) of the mice that received intravenous infusion of 1.0 × 10^5^ ADSCs (corresponding to 5.0 × 10^6^ cells per kilogram body weight) died immediately following the infusion. As shown in Figure 2, histological examination revealed no thrombus in the PBS group and CD41-staining was negative. In all of the mice infused with ADSCs, multiple pulmonary thrombi were observed, and these thrombi were positive for CD41. Meanwhile, few pulmonary thrombi were detected in mice infused with BMSCs, and CD41-staining was negative. These findings indicated that the infusion of high-dose ADSCs resulted in a higher incidence of death caused by pulmonary embolism than BMSCs. 

### 2.3. Comparison of the Procoagulation Factors and Humoral Factors Involved in the Therapeutic Effects of ADSCs and BMSCs

According to previous studies, the procoagulant activity of MSCs is induced by the expression of TF on their surface [16,17]. Plasminogen activator inhibitor-1 (PAI-1) is a powerful antifibrinolytic protein and is reportedly associated with an increased risk of thrombotic occlusive vascular disease [18]. Thus, we compared TF and PAI-1 mRNA levels between ADSCs and BMSCs. As shown in Figure 3a, the TF mRNA level was significantly higher in ADSCs than in BMSCs, while the PAI-1 mRNA level was not significantly different between ADSCs and BMSCs. Next, we investigated the mRNA expression levels of PTGES, TSG-6 and VEGF, all of which are secreted from MSCs and are involved in their anti-inflammatory, antifibrotic and angiogenic effects [13,19,20,21]. We found that the levels of TSG-6 and PTGES mRNA were upregulated in ADSCs compared with those in BMSCs, whereas the level of VEGF mRNA was upregulated in BMSCs compared with that in ADSCs (Figure 3b).

### 2.4. Comparison of the Therapeutic Effects of ADSCs and BMSCs Cultured in Serum-Free Medium on Renal Fibrosis in IRI Rats

Next, we compared the therapeutic effects of ADSCs and BMSCs cultured in serum-free medium (SF-ADSCs and SF-BMSCs, respectively) on renal fibrosis in IRI rats. As shown in Figure 4a, the protein levels of α-SMA and TGF-β1 were remarkably increased in the PBS group compared with those in the sham group. These increases were significantly inhibited in IRI rats infused with either SF-ADSCs or SF-BMSCs, with no significant difference between the SF-ADSC and SF-BMSC groups. Immunostaining revealed that the α-SMA-, collagen type I—and collagen type III—positive areas were equally attenuated in the SF-ADSC and SF-BMSC groups, with no significant difference between the two (Figure 4b,c). Similarly, HE and Masson’s trichrome staining demonstrated that the tubulointerstitial injury score and the area of interstitial fibrosis was significantly reduced by treatment with either SF-ADSCs or SF-BMSCs (Figure 4d,e). These results indicated that SF-ADSCs and SF-BMSCs exert similar therapeutic effects on renal fibrosis.

### 2.5. Comparison of Tail Vein Infusion of SF-ADSCs and SF-BMSCs on the Incidence of Death in Mice

Next, we evaluated the risk of pulmonary embolisms induced by infusion of SF-ADSCs or SF-BMSCs into the tail vein of C57/BL6 mice. As shown in Table 2, all of the mice (n = 10/10) that received PBS, 1.0 × 10^5^, 2.0 × 10^5^ or 3.0 × 10^5^ SF-ADSCs were alive at 24 h post infusion. By contrast, 90% (n = 9/10) of the mice that received a high-dose infusion of SF-ADSCs (4.0 × 10^5^ cells per mouse, corresponding to 2.0 × 10^7^ cells per kilogram body weight) died immediately following infusion. When the same high dose of SF-BMSCs (4.0 × 10^5^ cells) was infused, all of the mice (n = 10/10) were alive 24 h later. As shown in Figure 5, histopathological analysis revealed no thrombi in the PBS group, and CD41-staining was negative. The mice receiving a high-dose infusion of 4.0 × 10^5^ SF-ADSCs had massive pulmonary thrombi and these thrombi are positive for CD41. On the contrary, a few pulmonary thrombi were found in mice infused with high dose of SF-BMSCs, and SF-BMSC groups are slightly positive for CD41. These findings demonstrated that SF-ADSCs infused at a high dose led to a higher incidence of death caused by pulmonary embolism than SF-BMSCs.

### 2.6. Comparison of the Procoagulation Factors and Humoral Factors Involved in the Therapeutic Effects of SF-ADSCs and SF-BMSCs

We also compared TF and PAI-1 mRNA levels between SF-ADSCs and SF-BMSCs. As shown in Figure 6a, the TF mRNA level was significantly higher in SF-ADSCs than in SF-BMSCs. There was no significant difference in the PAI-1 mRNA level between SF-ADSCs and SF-BMSCs. Furthermore, we examined the mRNA expression levels of PTGES, TSG-6 and VEGF in SF-ADSCs and SF-BMSCs. While the mRNA levels of TSG-6 and PTGES were upregulated in SF-ADSCs compared with those in SF-BMSCs, the mRNA level of VEGF was upregulated in SF-BMSCs compared with that in SF-ADSCs (Figure 6b).

## 3. Discussion

In this study, we directly compared the antifibrotic effects of ADSCs and BMSCs derived from the same rats. Our study clarified that ADSCs attenuated IRI-induced renal fibrosis in rats more robustly than BMSCs when cultured in serum-containing medium. By contrast, ADSCs and BMSCs cultured in serum-free medium suppressed renal fibrosis equally well in IRI rats. The risk of pulmonary embolism was higher with ADSCs than with BMSCs, regardless of the presence of serum in the culture medium, indicating that attention should be given to the risk of embolism when using ADSCs. MSCs are expected to be developed as a new therapeutic tool to prevent progression from kidney injury to failure [22,23]. The present study provides evidence to support physicians in choosing between ADSCs and BMSCs in the treatment of kidney disease.

We have previously reported that serum-free medium promotes the immunosuppressive ability of BMSCs, thereby enhancing their antifibrotic effect. In this study, although ADSCs exerted greater therapeutic effects than BMSCs when cultured in serum-containing medium, this difference in therapeutic efficacy disappeared when serum-free medium was used. This might suggest that the degree of enhancement of the antifibrotic effects by serum-free conditions differs depending on the tissue from which the MSCs were derived. Additionally, under both serum-containing and serum-free conditions, the mRNA expression levels of TSG-6 and PTGES were higher in ADSCs than in BMSCs, and the VEGF mRNA level was higher in BMSCs than in ADSCs. These findings suggest that the infusion of ADSCs may be more suitable for anti-inflammatory therapy, while the infusion of BMSCs may be more beneficial for angiogenic therapy. Indeed, previous studies have reported that ADSCs have greater therapeutic effects than BMSCs in animal models of cisplatin-induced acute kidney injury [24] and acute liver failure [25]. Additionally, a meta-analysis of 11 studies of osteoarthritis showed that the therapeutic effects ADSCs are more stable than those of BMSCs [26]. Meanwhile, BMSCs were demonstrated to have a stronger therapeutic efficacy than ADSCs in models of ischemic stroke [27,28] and lower limb ischemia [29]. Taken together, the above-mentioned studies and our present results suggest that, when treating patients, both the target disease and culture method should be considered in deciding between ADSCs and BMSCs.

Previous studies have highlighted the risk of thrombogenic events following MSC infusions. In animals, high-dose MSC administration has reportedly caused severe symptoms, such as respiratory and circulatory failure [15,30]. In clinical trials, pulmonary embolisms have been reported among patients who received autologous ADSCs [31]. The procoagulant activity of MSCs, which is considered to be the mechanism underlying these embolisms, has been demonstrated by several studies to involve TF expression on the MSC surface. TF initiates blood coagulation, which could conceivably induce pulmonary embolisms [32,33]. To our knowledge, this is the first study that compares the risk of pulmonary embolism involved in the infusion of ADSCs and BMSCs cultured in serum-free medium as well as serum-containing medium. We found that the number of deaths from pulmonary embolism was higher in mice infused with ADSCs than with BMSCs, regardless of the presence of serum in the culture medium. Moreover, we also showed that the expression level of TF was significantly higher in ADSCs than in BMSCs, regardless of the presence of serum in the culture medium. We suspect that the higher occurrence of embolism with ADSCs is attributable to this procoagulant factor. Therefore, thromboembolism should be considered in the clinical use of ADSCs.

A major limitation of this study is that we were unable to collect data on the antifibrotic abilities of human MSCs. This is because of challenges and ethical issues associated with the procurement of adipose tissue and bone marrow from the same patient. Further studies to investigate the comparative antifibrotic effects of human ADSCs and BMSCs are warranted.

## 4. Materials and Methods

### 4.1. MSC Preparation

We collected bone marrow from Sprague–Dawley (SD) male rats at 6–8 weeks of age as described previously [13]. The cells were grown in DMEM (Sigma-Aldrich, St. Louis, MO, USA) with 10% FBS (Sigma-Aldrich) for primary and secondary culture, and the MSCs derived from bone marrow were cryopreserved. After the collection of bone marrow, anterior subcutaneous adipose tissue was obtained from the rats and washed with PBS. After crushing the tissue into small pieces under the laminar hood, the cell pellet was cultured (without the use of collagenase). After two passages, the resulting MSCs derived from adipose tissue were cryopreserved. Next, the MSCs derived from both tissues were cultured under 10% DMEM or STK2 (Kanto Chemical, Tokyo, Japan), a serum-free medium containing growth factors for MSCs. The resulting cells at Passage 4 grown in DMEM were designated as ADSCs and BMSCs, while those grown in STK2 were designated as SF-ADSCs and SF-BMSCs. These cells were all cultured at 37 °C in humidified air containing 5% CO_2_ and counted using a LUNA-FX7™ (Logos Biosystems, Anyang, Gyeonggi-do, Republic of Korea) before passaging or experiments.

### 4.2. Characterization of MSCs

We validated the quality of ADSCs and BMSCs using flow cytometry. As shown in Appendix A online, the cells were positive for the standard MSC markers CD44 and CD90 and negative for CD45.

### 4.3. Animals

We used male SD rats and male C57/BL6 mice purchased from Charles River Laboratories Japan (Yokohama, Japan). Rats were induced with IRI at 8 weeks of age. Mice were used to examine the risk of pulmonary embolism induced by MSCs at 7 weeks of age. All experimental procedures were approved by the Institutional Animal Care and Use Committee of Hiroshima University (Hiroshima, Japan; Permit Number, A18-174-2 for rats and A22-31 for mice) and conducted in accordance with the Guide for the Care and Use of Laboratory Animals, 8th ed., 2010 (National Institutes of Health, Bethesda, MD, USA). This study is reported in accordance with the ARRIVE guidelines.

### 4.4. Experimental Animal Models

To establish the IRI model, SD rats were randomly divided into six groups (n = 5 in each group): sham, PBS (control), ADSC, BMSC, SF-ADSC and SF-BMSC. Rats were anesthetized by intraperitoneal infusion of a mixture of three anesthetic agents: medetomidine, midazolam and butorphanol. After a laparotomy was performed, the left renal pedicle was clamped using atraumatic vascular clamps for 1 h, followed by reperfusion on a heating blanket. After reperfusion, a 0.2 mL mixture of PBS, or ADSCs, BMSCs, SF-ADSCs and SF-BMSCs (2.5 × 10^5^ cells per rat) diluted in PBS was infused through the abdominal aorta, which was clamped above and below the left renal artery bifurcation using 32-gauge needles. At 21 days post infusion, the rats were sacrificed, and their kidneys were collected. 

To investigate the risk of MSC-induced pulmonary embolism, C57/BL6 mice were randomly divided into five groups (n = 6–10 in each group): PBS (control), ADSC, BMSC, SF-ADSC and SF-BMSC. Mice were anesthetized by intraperitoneal infusion of the mixture of three anesthetic agents described above. Using a 29-gauge needle, a 0.2 mL solution of PBS or a mixture of ADSCs, BMSCs, SF-ADSCs and SF-BMSCs suspended in PBS (range, 1.0 × 10^5^ to 4.0 × 10^5^ cells per mouse) was infused into C57/BL6 mice via the tail vein. The lungs were immediately isolated from mice that died from the infusion. Mice that survived the infusion were sacrificed 1 day later, after which the lungs were isolated.

### 4.5. Histological Analysis

Formalin-fixed, paraffin-embedded (FFPE) blocks of the kidneys of rats collected at 21 days post renal IRI were prepared. Sections (2 µm thickness) of the FFPE kidney tissues were stained with HE and Masson’s trichrome to evaluate histological injury and the extent of interstitial fibrosis. For the evaluation of tubulointerstitial injury, the extent of damage to the renal cortex was graded on a scale of 0 to 4 as previously described [34] using the average of ten randomly selected fields (×200) of the cortex on HE. For the assessment of the areas of interstitial fibrosis, we analyze by examining the average of five randomly selected fields (×200) of the cortex on Masson staining. Each sample were observed using a Lumina Vision (Mitani, Osaka, Japan). FFPE tissue sections were also prepared from the lungs of mice that were collected following infusion or sacrifice. FFPE lung sections (4 µm thickness) were stained with hematoxylin and eosin to assess pulmonary emboli.

### 4.6. Immunohistochemical Analysis

Immunohistochemical staining of the rat FFPE kidney sections (4 μm thickness) was performed as previously described [13]. The following primary antibodies were used: mouse monoclonal α-SMA antibody (Sigma-Aldrich), anti-collagen type I antibody (Bio-Rad, Hercules, CA, USA) and anti-collagen type III antibody (abcam, Cambridge, UK). Immunostaining for α-SMA was performed with the EnVision System (Dako, Glostrup, Denmark). Collagen type I and collagen type III were identified with the ABC system (Vector Laboratories, Newark, CA, USA). The positive areas for α-SMA, collagen type I and collagen type III staining were assessed from the average of five randomly selected fields (×200) for each rat using ImageJ software 1.53 (National Institutes of Health). Immunohistochemical staining for CD41(abcam) was performed on mouse FFPE lung sections (2 μm thickness). CD41 was identified using the ABC system.

### 4.7. Western Blotting

Sample collection and Western blotting were performed as previously described [13]. The following primary antibodies were used: mouse monoclonal anti- glyceraldehyde-3-phosphate dehydrogenase (GAPDH) antibody (Sigma-Aldrich), mouse monoclonal anti-α-SMA antibody (Sigma-Aldrich) and mouse monoclonal TGF-β1 antibody (abcam). The following secondary antibodies were used: horseradish-peroxidase-conjugated goat anti-rabbit immunoglobulin G (Dako) or goat anti-mouse immunoglobulin G (Dako). The SuperSignal West Pico system (Thermo Fisher Scientific, Rockford, IL, USA) or ImmunoStar LD (FUJIFILM Wako Pure Chemical Corporation, Osaka, Japan) was used to detect signals. The intensity of each band was analyzed using ImageJ software 1.53 and normalized to the level of GAPDH.

### 4.8. Quantitative Real-Time Reverse Transcription PCR

RNA extraction and real-time reverse transcription PCR were performed using previously described methods [13]. Specific oligonucleotide primers and probes for rat TF (assay ID: Rn00564925_m1), rat PAI-1 (assay ID: Rn01481341_m1), rat PTGES (assay ID: Rn00572047_m1), rat TSG-6 (assay ID: Rn01753871_m1) and rat GAPDH (assay ID: Rn01775763_g1) were obtained as TaqMan Gene Expression Assays (Applied Biosystems, Foster City, CA, USA). The levels of mRNA were normalized to the level of GAPDH.

### 4.9. Statistical Analysis

All data are expressed as the mean ± standard deviation. Statistical analyses were performed using a one-way analysis of variance followed by Tukey’s post-hoc testing. Differences between two groups were analyzed by Student’s *t* test. A *p* value < 0.05 was defined as indicating statistical significance. All analyses were carried out with JMP^®^ Pro 16.2.0 software (SAS Institute Inc., Cary, NC, USA).

## 5. Conclusions

In conclusion, ADSCs had a stronger suppressive effect on IRI-induced renal fibrosis than BMSCs, but only when cultured in serum-containing medium; ADSCs and BMSCs cultured in serum-free medium suppressed renal fibrosis equally well. In clinical applications, ADSCs are more effective than BMSCs for diseases associated with inflammation and fibrosis when cultured in serum-containing medium. When using MSCs cultured in serum-free medium, ADSCs and BMSCs are expected to have the same therapeutic effect for these diseases. Our results also pointed out the need for caution regarding the risk of thromboembolism when using ADSCs. Given the increasing popularity of MSC-based therapy, these findings suggest that we should decide between the use of ADSCs and BMSCs in patients based on both the target disease and the MSC culture method.

## Figures and Tables

**Figure 1 ijms-24-16920-f001:**
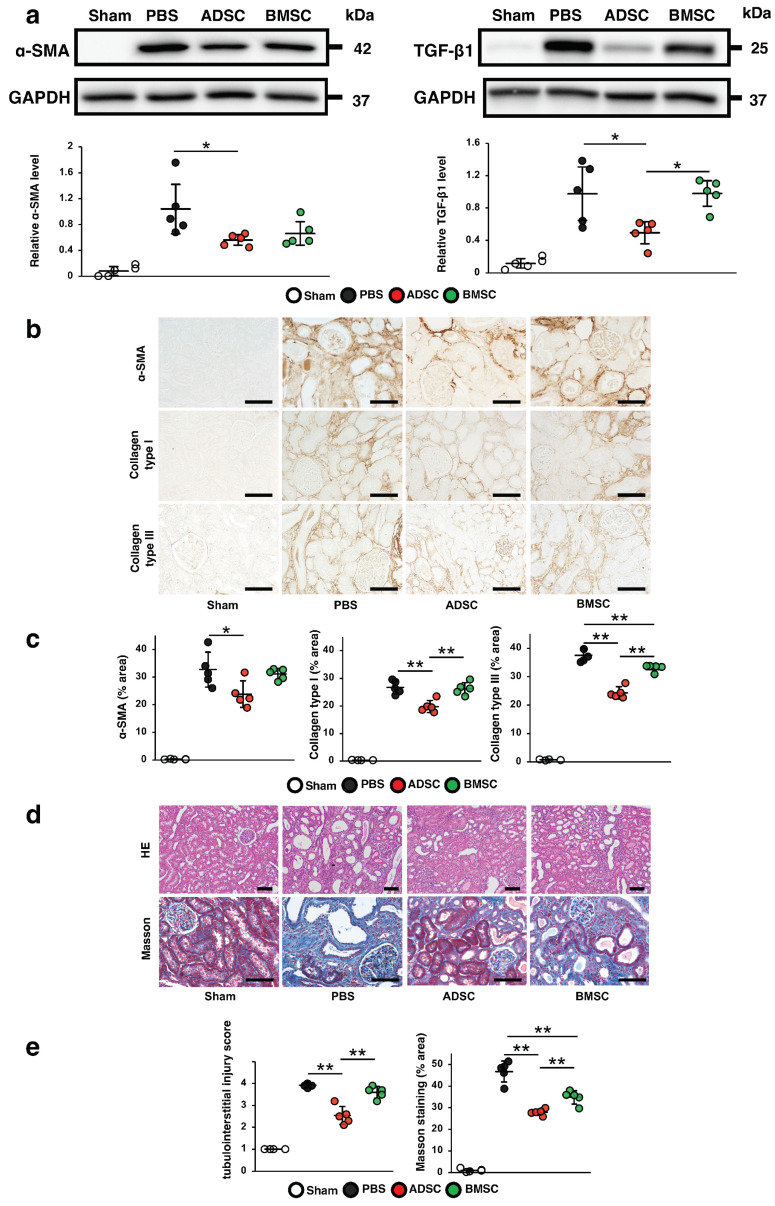
Antifibrotic effects of infusion of ADSCs or BMSCs at 21 days post ischemia-reperfusion injury (IRI) in rats. (**a**) Western blotting analysis of the fibrosis markers alpha-smooth muscle actin (α-SMA) and transforming growth factor (TGF)-β1 in the kidney cortex. Graphs show densitometric analyses of α-SMA and TGF-β1 expression levels normalized to glyceraldehyde-3-phosphate dehydrogenase (GAPDH). (**b**) Representative immunohistochemical staining of α-SMA, and collagen types I and III in kidney sections (scale bar = 100 μm). (**c**) Quantification of α-SMA- and collagen types I- and III-positive areas as percentages of the total area (five sections analyzed per animal). (**d**) Representative image of hematoxylin and eosin (HE) staining and Masson’s trichrome staining in kidney sections (scale bar = 100 µm). (**e**) Quantification of tubulointerstitial injury by HE staining (ten sections analyzed per animal) and area of interstitial fibrosis as a percentage of the total area of Masson’s trichrome staining (five sections analyzed per animal). Data are expressed as means ± standard deviation; * *p* < 0.05, ** *p* < 0.01. One-way analysis of variance followed by Tukey’s post-hoc testing was used to verify differences between the models. ADSC: mesenchymal stem cells derived from adipose tissue; BMSC: mesenchymal stem cells derived from bone marrow; PBS: phosphate-buffered saline group.

**Figure 2 ijms-24-16920-f002:**
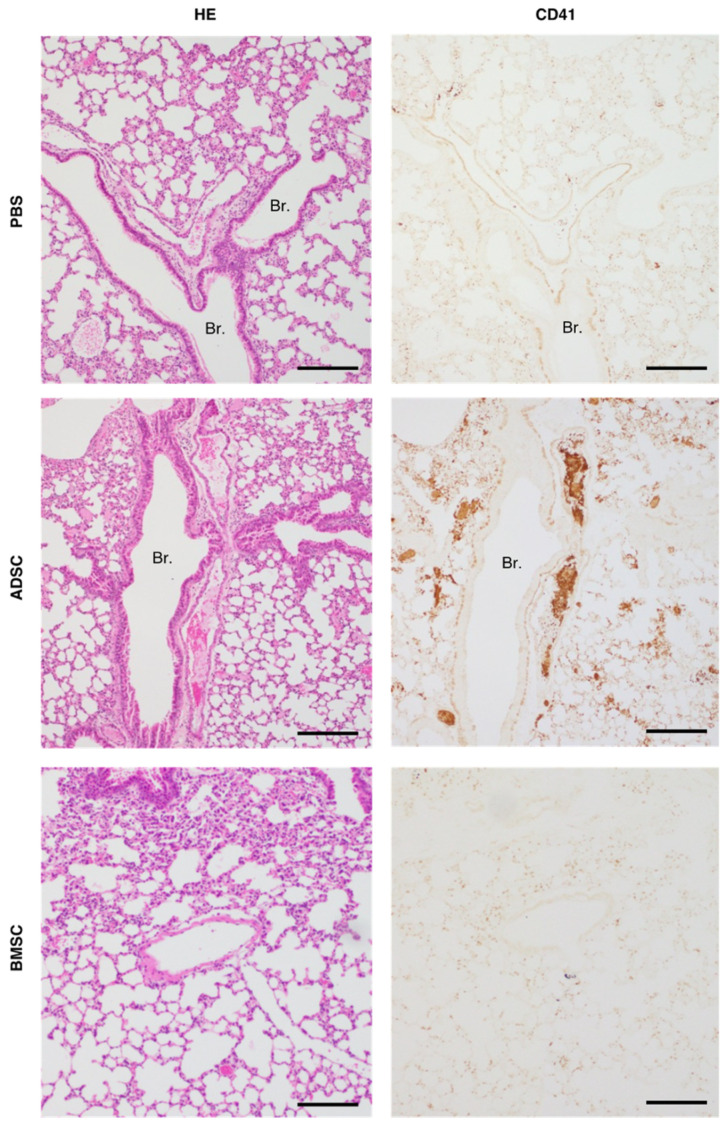
Histopathological analysis of the lung in mice receiving intravenous infusion with PBS, rat ADSCs or BMSCs. Representative histological images of hematoxylin and eosin-stained lung sections from mice infused with PBS (sacrificed 24 h after injection), 1.0 × 10^5^ ADSC (died immediately after injection), or 1.0 × 10^5^ BMSCs (sacrificed 24 h after injection) via the tail vein (scale bar = 200 μm). The photos of CD41-staining above show the same kidney sections (scale bar = 200 µm). The micrograph of the lungs of mice injected with 1.0 × 10^5^ ADSC shows thrombi in the blood vessels. Br.: bronchiole. Other abbreviations are shown in Figure 1.

**Figure 3 ijms-24-16920-f003:**
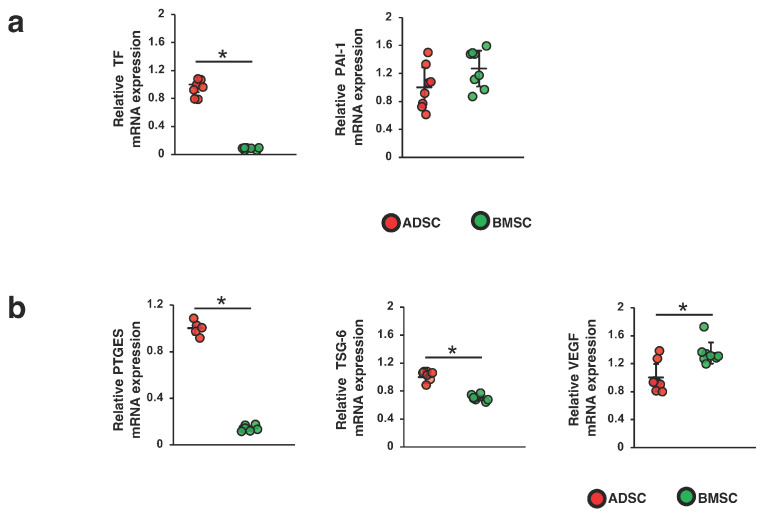
Expression levels of procoagulation and humoral factors involved in therapeutic effects in ADSCs and BMSCs. Expression levels of (**a**) tissue factor (TF) and plasminogen activator inhibitor-1 (PAI-1) mRNAs and (**b**) prostaglandin E synthase (PTGES), tumor necrosis factor-α-induced protein 6 (TSG-6) and vascular endothelial growth factor (VEGF) mRNAs in ADSCs and BMSCs cultured in serum-containing medium. Differences between two groups were analyzed by Student’s *t* test; * *p* < 0.05. Abbreviations are shown in Figure 1.

**Figure 4 ijms-24-16920-f004:**
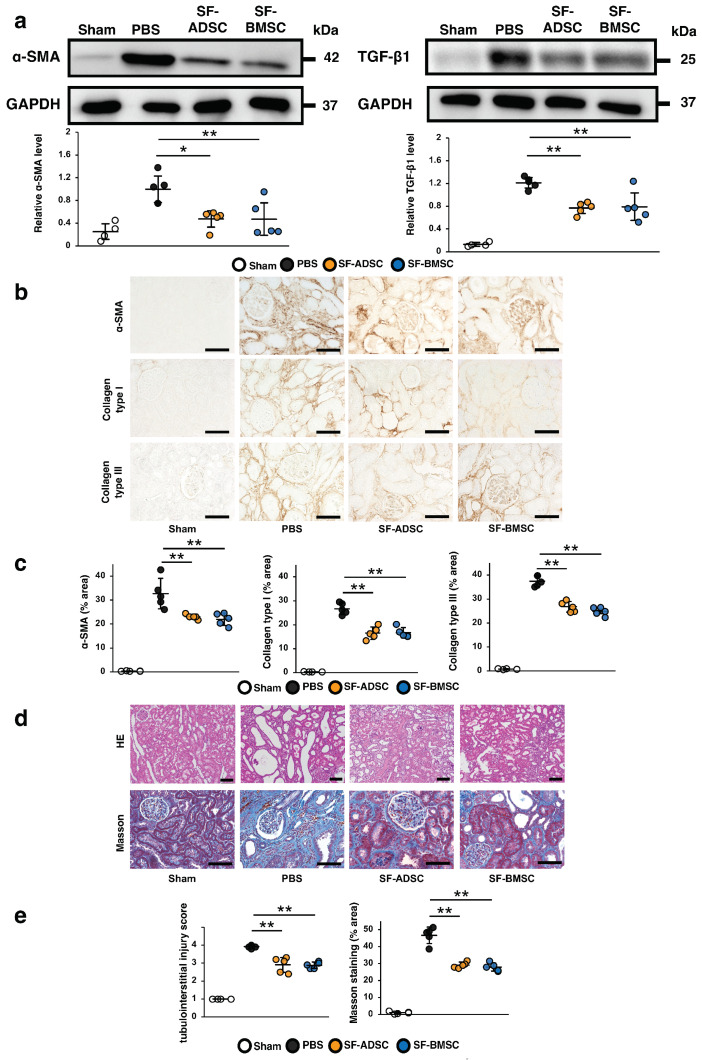
Antifibrotic effects of ADSCs and BMSCs cultured in serum-free medium at 21 days post ischemia-reperfusion injury (IRI) in rats. (**a**) Western blotting analysis of the fibrosis markers alpha-smooth muscle actin (α-SMA) and transforming growth factor (TGF)-β1 in the kidney cortex. Graphs show densitometric analyses of α-SMA and TGF-β1 expression levels normalized to glyceraldehyde-3-phosphate dehydrogenase (GAPDH). (**b**) Representative immunohistochemical staining of α-SMA, and collagen types I and III in kidney sections (scale bar = 100 μm). (**c**) Quantification of α-SMA- and collagen types I- and III-positive areas as percentages of the total area (five sections analyzed per animal). (**d**) Representative image of hematoxylin and eosin (HE) staining and Masson’s trichrome staining in kidney sections (scale bar = 100 µm). (**e**) Quantification of tubulointerstitial injury by HE staining (ten sections analyzed per animal) and area of interstitial fibrosis as a percentage of the total area of Masson’s trichrome staining (five sections analyzed per animal). Data are expressed as means ± standard deviation; * *p* < 0.05, ** *p* < 0.01. One-way analysis of variance followed by Tukey’s post-hoc testing used to verify differences between the models. SF-ADSC mesenchymal stem cells derived from adipose tissue cultured in serum-free medium; SF-BMSC mesenchymal stem cells derived from bone marrow cultured in serum-free medium; PBS phosphate-buffered saline.

**Figure 5 ijms-24-16920-f005:**
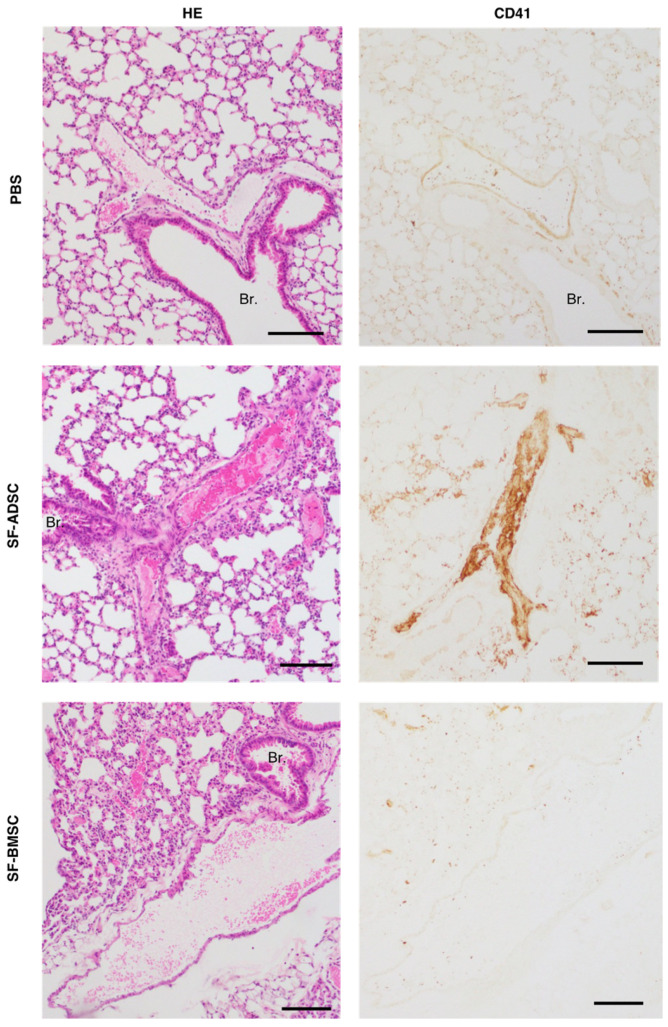
Histopathological analysis of the lung in mice receiving intravenous infusion with PBS, rat ADSCs or BMSCs cultured in serum-free medium. Representative histological images of hematoxylin- and eosin-stained lung sections from mice infused with PBS (sacrificed 24 h after injection), 4.0 × 10^5^ SF-ADSC (died immediately after injection), or 4.0 × 10^5^ SF-BMSCs (sacrificed 24 h after injection) via the tail vein (scale bar = 200 μm). The photos of CD41-staining above show the same kidney sections (scale bar = 200 µm). The micrograph of the lungs of mice injected with 4.0 × 10^5^ SF-ADSC shows massive thrombi in the blood vessels. Br.: bronchiole. Other abbreviations are shown in Figure 4.

**Figure 6 ijms-24-16920-f006:**
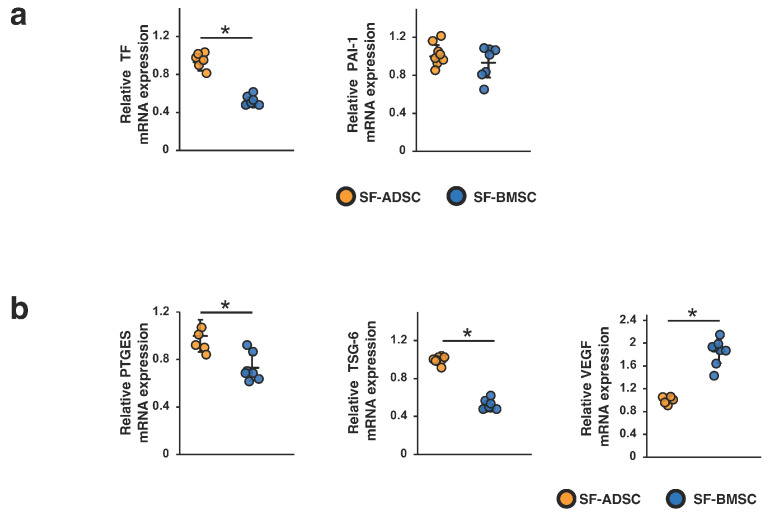
Expression level of procoagulation and humoral factors involved in therapeutic effects in ADSCs and BMSCs in serum-free medium. Expression levels of (**a**) tissue factor (TF) and plasminogen activator inhibitor-1 (PAI-1) mRNAs and (**b**) prostaglandin E synthase (PTGES), tumor necrosis factor-α-induced protein 6 (TSG-6) and vascular endothelial growth factor (VEGF) mRNAs in ADSCs and BMSCs cultured in serum-free medium. Differences between two groups were analyzed by Student’s *t* test; * *p* < 0.05. Abbreviations are shown in Figure 4.

**Table 1 ijms-24-16920-t001:** Survival rate of mice post infusion of ADSCs and BMSCs via the tail vein.

Number of Injected Cells	Total Mouse Number	Number of Surviving Mice	Survival Rate
1.0 × 10^5^ ADSCs/mouse	10	6	60%
1.0 × 10^5^ BMSCs/mouse	8	8	100%
0 (PBS, the same volume)	6	6	100%

ADSC mesenchymal stem cells derived from adipose tissue cultured in serum-free medium; BMSC mesenchymal stem cells derived from bone marrow cultured in serum-free medium; PBS phosphate-buffered saline.

**Table 2 ijms-24-16920-t002:** Survival rate of mice post infusion of SF-ADSCs and SF-BMSCs via the tail vein.

Number of Injected Cells	Total Mouse Number	Number of Surviving Mice	Survival Rate
1.0 × 10^5^ SF-ADSCs/mouse	10	10	100%
2.0 × 10^5^ SF-ADSCs/mouse	6	6	100%
3.0 × 10^5^ SF-ADSCs/mouse	10	10	100%
4.0 × 10^5^ SF-ADSCs/mouse	10	1	10%
4.0 × 10^5^ SF-BMSCs/mouse	10	9	90%
0 (PBS, the same volume)	6	6	100%

SF-ADSC: mesenchymal stem cells derived from adipose tissue cultured in serum-free medium; SF-BMSC: mesenchymal stem cells derived from bone marrow cultured in serum-free medium; PBS: phosphate-buffered saline.

## Data Availability

The data that support the findings of this study are available from the corresponding author upon reasonable request.

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
