# Peer review of "Comparison of the Therapeutic Effects of Adipose- and Bone Marrow-Derived Mesenchymal Stem Cells on Renal Fibrosis"

_ijms, 2023, doi:10.3390/ijms242316920_

Round 1
Reviewer 1 Report
Comments and Suggestions for Authors
The present manuscript compares the therapeutic effects of adipose- and bone 2 marrow-derived mesenchymal stem cells on renal fibrosis, a every interesting subject with impact in chronic kidney disease research area. The title is clear and easy to be individualized. The abstract is well structured. The Introduction is well documented; but some more data about the role of adipose- and bone narrow derived mesenchymal stem cells in renal fibrosis shoukd be presented. Discussion are well conceived and support the results. The impact on clinical research should be largely presented. The Conclusions should be clearly present for stating the idea of the title.
Comments on the Quality of English LanguageMinor writting errors should be corrected
Reviewer 2 Report
Comments and Suggestions for Authors
This study compares the antifibrotic effects of adipose-derived MSCs (ADSCs) and bone marrow-derived MSCs (BMSCs) from the same rats. When cultured in serum-containing medium, ADSCs show a more potent inhibitory effect on renal fibrosis induced by ischemia-reperfusion injury in rats compared to BMSCs. However, both ADSCs and BMSCs cultured in serum-free medium are equally effective in suppressing renal fibrosis. The research highlights the importance of choosing between ADSCs and BMSCs based on the target disease and culture method, considering thromboembolism risks in clinical applications.
Comments and questions:
- The color code represented in the graph legend is not applied in the graphs represented by the authors.
- In the kidney sections (figure 1 and 4) of rats treated with ADSC and BMSC, the representative immunostaining figures are not clear and it seems that there was a bubble that inhibited the full staining which can effect the quantification. Can the authors comment on this?
- The representative figures of immunostaining of the sham group should be changed because they do not show clear sections of the kidney. It is like they represent a background removal of an image or that the staining was not effective at all. Can the authors comment on this?
- Does the same outcome occur in regards to thrombus when rats are injected with a lower concentration with ADSCs?
- Why there was no decrease of PAI-1 levels in rats treated with BMSCs, knowing that they did not show any signs of embolism based on the immunostaining figures that the authors presented in figure 2?
- With the lower concentrations of SF-ADSCs, did the authors saw a reduction of embolism in the rats?
- Did the authors tried the use of secretomes and/or EVs secreted from these stem cells and saw their effect on renal fibrosis and pulmonary embolism?
